# Breaking a dative bond with mechanical forces

Pengcheng Chen[1,9], Dingxin Fan[2,9], Yunlong Zhang [3✉], Annabella Selloni [4], Emily A. Carter [5,6], Craig B. Arnold [1,5], David C. Dankworth [3], Steven P. Rucker [3], James R. Chelikowsky [2,7,8✉] & Nan Yao [1✉]

Bond breaking and forming are essential components of chemical reactions. Recently, the structure and formation of covalent bonds in single molecules have been studied by non-contact atomic force microscopy (AFM). Here, we report the details of a single dative bond breaking process using non-contact AFM. The dative bond between carbon monoxide and ferrous phthalocyanine was ruptured via mechanical forces applied by atomic force microscope tips; the process was quantitatively measured and characterized both experimentally and via quantum-based simulations. Our results show that the bond can be ruptured either by applying an attractive force of ~150 pN or by a repulsive force of ~220 pN with a significant contribution of shear forces, accompanied by changes of the spin state of the system. Our combined experimental and computational studies provide a deeper understanding of the chemical bond breaking process.

[1] Princeton Institute for the Science and Technology of Materials, Princeton University, Princeton, NJ 08540-8211, USA. [2] McKetta Department of Chemical Engineering, University of Texas at Austin, Austin, TX 78712-1589, USA. [3] ExxonMobil Research and Engineering Company, Annandale, NJ 08801-3096, USA. [4] Department of Chemistry, Princeton University, Princeton, NJ 08544-0001, USA. [5] Department of Mechanical and Aerospace Engineering, Princeton University, Princeton, NJ 08544-5263, USA. [6] Office of the Chancellor and Department of Chemical and Biomolecular Engineering, University of California, Los Angeles, Los Angeles, CA 90095-1405, USA. [7] Department of Physics, University of Texas at Austin, Austin, TX 78712-1192, USA. [8] Center for Computational Materials, Oden Institute for Computational Engineering and Sciences, University of Texas at Austin, Austin, TX 78712-1229, USA. [9] These authors contributed equally: Pengcheng Chen, Dingxin Fan. ✉email: nyao@princeton.edu; yunlong.zhang@exxonmobil.com; jrc@utexas.edu

The ability to obtain images of organic molecules with atomic resolution was first demonstrated by Gross et al. in 2009 by using a carbon monoxide (CO) molecule attached to an Atomic Force Microscope (AFM) tip[1] mounted on a qPlus sensor[2]. This work inspired a wide range of applications, including directly characterizing molecular structures[3–5], probing molecular properties[6–10], creating new structures[11,12], and even providing a tool for studying various types of chemical bonds, such as hydrogen bonds and halogen bonds[13,14]. These studies stimulated significant discussions on the contrast mechanism of AFM images and on the extent to which the image could represent a physical description of a chemical bond[15,16]. A similar technique was used to directly manipulate individual chemical bonds. Wagner et al.[17] measured the binding energies (including nonspecific interactions) between an organic molecule, 3,4,9,10-perylene-teracarboxylic-dianhydride, and a metal substrate, Au(111). Recently, Kawai et al. measured the $C=O \cdots H\text{-}C$ bonding interaction between a CO tip and the C-H group of an aromatic hydrocarbon before the onset of Pauli repulsion[18]. Huber et al. studied the interactions between a CO tip with Fe, Cu, and Si adatoms and revealed the bond-forming process during the transition from physisorption to chemisorption[19].

These studies using AFM tips to manipulate chemical bonds provided insights into the bond-forming process involved in many surface interactions. However, the controlled breaking of a chemical bond using mechanical forces, along with accurate measurements of these forces, is also important and its detailed process has not yet been fully understood. Understanding the process of rupturing a bond is essential for obtaining insights into the physical nature of a chemical bond and its role in many chemical and catalytic mechanisms. Dative bonds are commonly found in transition metal complexes and play vital roles in catalysis, organometallic chemistry, and biochemistry. Here, we focus on understanding the breaking of a single chemical bond between a CO molecule and a ferrous phthalocyanine (FePc) complex using AFM together with real-space pseudopotential density functional theory (DFT) calculations[20–22]. Our results reveal detailed mechanisms of bond breaking by both repulsive and attractive forces. This work advances understanding of the origins of measured forces in dative bond breaking.

## Results

**STM/AFM topography and structure identification.** We prepared a supported CO-FePc system by dosing CO molecules onto FePc adsorbed on a Cu(111) surface at 4.8 K. The scanning tunneling microscope (STM) image in Fig. 1a shows two distinct features for the FePc molecule and the complex (CO-FePc), which are similar to previously reported STM images on other surfaces[23]. AFM images were obtained using a CO-terminated tip, confirming their respective structures (Fig. 1b, c). The AFM image in Fig. 1b of the CO-FePc complex featured a protruding center due to the CO attached to Fe. This characterization is confirmed by comparing with AFM images of FePc molecules on the surface Fig. 1c, and further verified by our simulated images (Fig. 1d, e).

**Breaking a dative bond with AFM tips.** The dative CO-FePc bond is known to be formed via σ-donation from the CO 5σ orbital and π-back donation from Fe dπ[24–27]. We studied the rupture of this dative bond by applying mechanical forces using the AFM tip. The same CO-terminated tip employed for imaging was used first because it is known to be chemically inert[28]. By decreasing the tip height, the repulsive interactions increased, as indicated by the increased contrast in the images (Fig. 2b–d). At a tip height of about +40 pm, the peripheral aromatic rings of FePc became visible, while the center of the image became distorted due to strong repulsions with the tip. Upon further reducing the tip height (+30 pm), a sudden change of the image occurred during scanning, as indicated by a line created with a different contrast. Subsequent scans showed the repulsion had disappeared, indicating that the CO attached to FePc was dislodged due to the strong repulsion with the tip. The chemical structure of FePc revealed from subsequent scanning of the lower part of the molecule confirmed that a free FePc was left after CO removal and that the tip remained intact during the dissociation. Comparison of the contrast in the lower part to the upper part of the same AFM image Fig. 2d obtained at the same tip height reveals a downward shift of FePc by ~30 pm upon CO removal. This shift indicates a trans effect by the Cu substrate on the FePc complex[24–27,29], whereby the binding or removal of one ligand

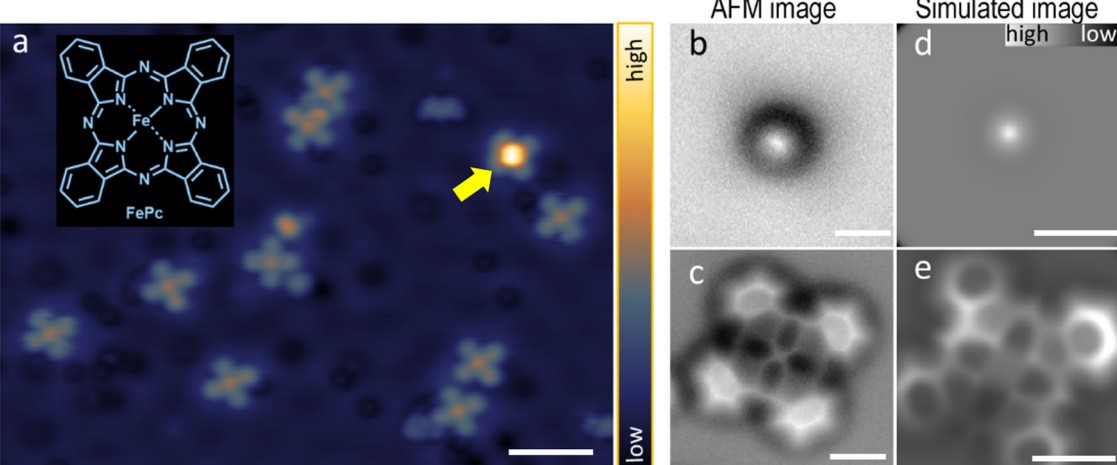

**Fig. 1 STM and AFM images of FePc and the datively bonded CO-FePc complex. a** STM image of the FePc molecule with (indicated by the arrow) and without adsorption of CO, with the insert showing the chemical structure of FePc (set point: $V_{sample} = +100$ mV, I = 100 pA, scale bar: 3 nm). **b, c** Experimental AFM images of FePc with and without adsorbed CO, obtained using a CO-terminated tip (V = 0 V, A = 100 pm, scale bar: 0.5 nm) at tip heights z of +160 pm and -10 pm, respectively. The tip height z was set with respect to a reference height given by the STM set point (100 mV, 100 pA) above the bare Cu(111) substrate in the vicinity of the molecule. The minus sign of tip height z indicates a decrease of tip height. **d, e** Simulated AFM images corresponding to CO-FePc and FePc at tip heights of 554 pm and 300 pm. The tip height in the simulation is defined as the distance between the front atom of the tip and the average height of the FePc complex (excluding the decorated CO) (scale bar: 0.5 nm).

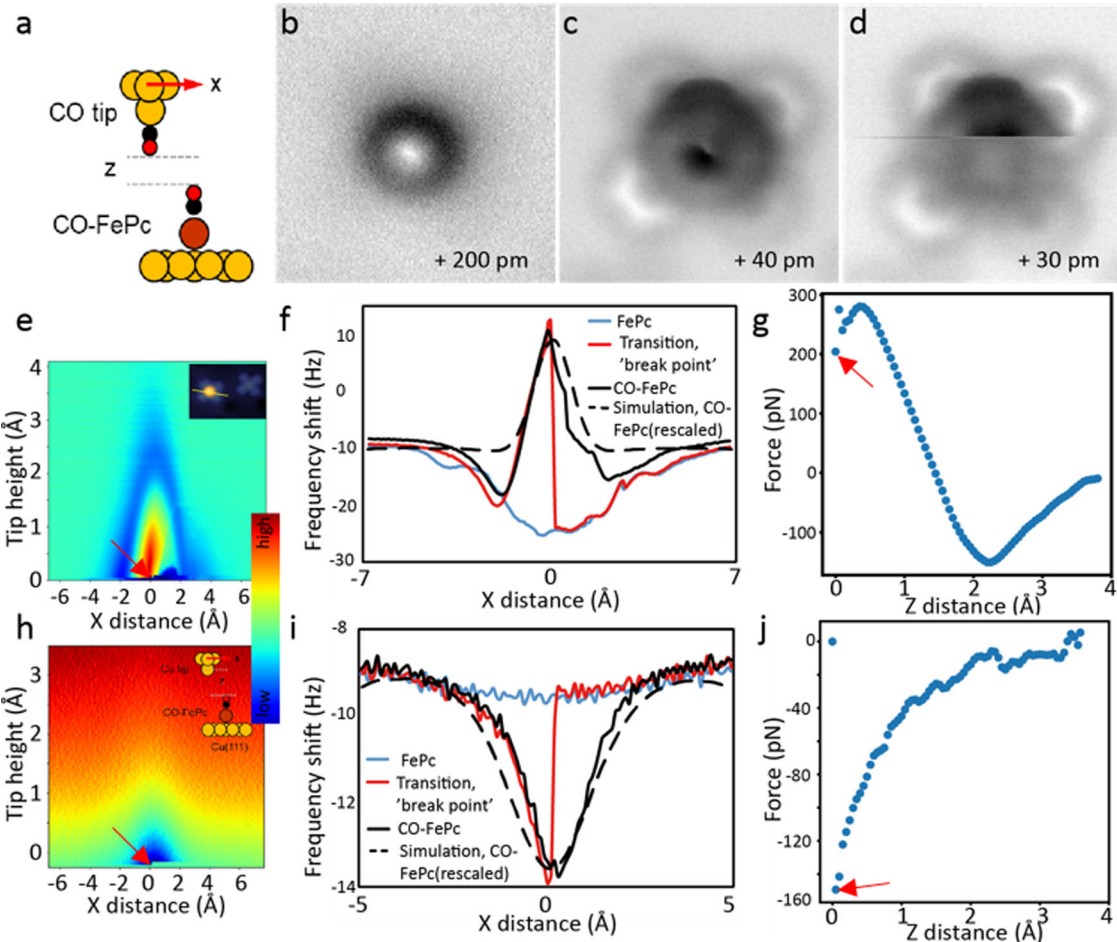

**Fig. 2 Rupturing the dative CO-FePc bond using AFM tips. a** Schematic of a CO-AFM tip interacting with CO-FePc (Cu: yellow; C: black; O: red; Fe: brown). **b–d** Non-contact AFM images obtained at different tip heights (z); the final dislodging of CO occurs at $z = +30$ pm. **e** 3D force map of the frequency shift ($\Delta f$) vs. AFM tip heights (z) and horizontal position (x), with a CO tip. Step size is 5 pm in z, and the scan path in x is across the center of the Fe, as shown in the inset. The tip position at bond rupture is indicated by the breakpoint (arrow). **f** Frequency shift ($\Delta f$) obtained in the horizontal (x) direction before, during (indicated by the disjointed curve), and after the bond rupture. **g** The force curve deconvoluted from $\Delta f$ at the breakpoint in the vertical (z) direction. **h** 3D force map of the frequency shift ($\Delta f$) showing quantitative rupture of the dative bond, obtained using a Cu tip; the insert shows schematic of interaction between a Cu tip and CO-FePc. **i** Frequency shift ($\Delta f$) obtained using a Cu tip scanned in the horizontal (x) direction. **j** The deconvoluted force curve at the breakpoint in the vertical (z) direction using a Cu tip. (Red arrows indicate the bond rupture point. Long-range background forces are subtracted in Figures **g** and **j**).

respectively reduces or enhances the strength of the bond to the ligand on the opposite side[30]. This observation confirms the rupture of the dative bond between the CO and FePc induced by the increased interactions during tip scanning.

To elucidate CO-FePc bond rupture, we performed detailed measurements of the interaction forces during the entire bond rupture process. Figure 2e shows a 3D force map representing the frequency shift ($\Delta f$) obtained at different tip heights (z) by scanning across the center of the CO-FePc complex (shown in the inset). The dislodging of the CO was indicated by a breakpoint ($x = 0$) with decreasing the tip height during scanning and by a discontinuity in the frequency shift ($\Delta f$) curve (red curve in Fig. 2f). The interaction force and energy were calculated from the measured frequency shift with the method and formulas proposed by Sader[31]. The force curve along with the tip height (z) at the breaking point ($x = 0$), in Fig. 2g, shows that the dative bond ruptured with a force of $220 \pm 30$ pN, after passing a maximal force at ~300 pN.

Dislodging experiments were also performed with a bare metal tip, which was terminated by a Cu atom under similar experimental conditions as for the CO-terminated tip. The Cu

atom tip is known to be a chemically active tip[32]. When we used the Cu tip, only attractive interactions between the tip and CO-FePc were detected (Fig. 2h), until the rupture of the dative bond took place. At this point, the attractive force reached $150 \pm 30$ pN by reducing the tip height (Fig. 2i, j).

**Real-space DFT calculations**. The surprising observation that both an attractive force of $-150$ pN and a repulsive force of $+220$ pN are capable of breaking the same dative bond highlights the important role of probe tips, although this result is consistent with findings by Berwanger et al. that the CO-terminated tip can exert forces of up to 450 pN without breaking off[33,34]. We employed real-space DFT calculations to address the role of the AFM tip and to shed light on the details of the bond-breaking process.

To understand interactions between the AFM tips and CO-FePc before the bond is broken, we computed frequency shifts of the probe tips at relatively large heights (z~5 Å) using optimized geometries[22] (Supplementary Figure 1). We modeled the Cu tip using a $Cu_2$ cluster and the CO tip using a Cu-CO cluster. We tested more complex tip conformations and found the effect in

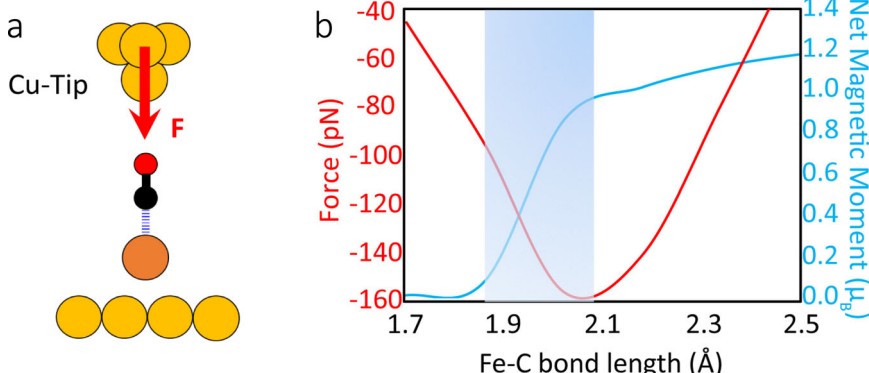

**Fig. 3 Real-space pseudopotential DFT calculations for the breaking of the dative bond in CO-FePc with a Cu tip. a** Schematic showing the interaction of the Cu tip with the CO-FePc complex on Cu(111) (Cu: yellow; C: black; O: red; Fe: brown). The red arrow indicates the attractive force acting on the tip apex. **b** The red curve shows the calculated attractive vertical force on the Cu apex while the blue curve shows the net magnetic moment as a function of the Fe-C bond length. The shaded area indicates where the bond rupture process occurred.

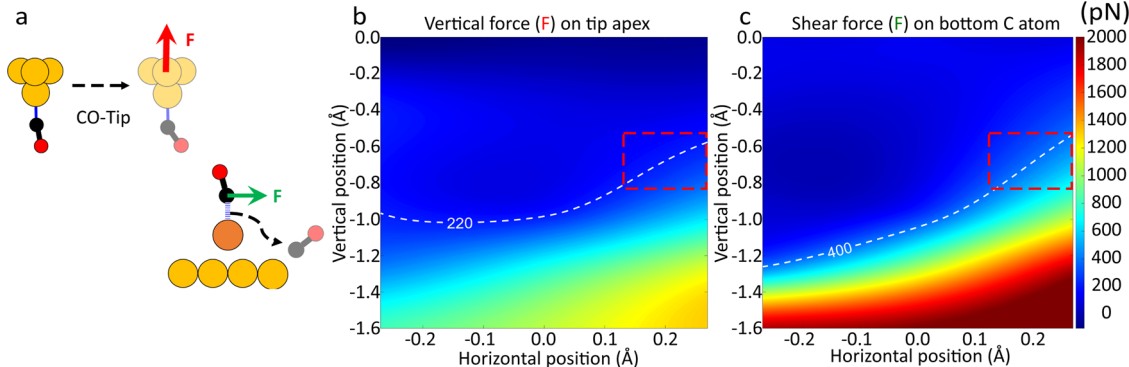

**Fig. 4 Real-space pseudopotential DFT calculations for the breaking of the dative bond in CO-FePc with a CO tip. a** Schematic shows the interactions between the CO tip and the CO-FePc complex on Cu(111). The red and green arrows indicate the direction of the forces acting on the tip apex and on the bottom C atom, respectively. (Cu: yellow; C: black; O: red; Fe: brown) **b, c** are line cuts of 3D force map for the calculated compressive vertical force on the tip apex, and the lateral force acting on the bottom C (of the CO attached to FePc). The x and y axes are the horizontal positions and heights of the tip with respect to an equilibrium position. The dashed curves in **b** and **c** correspond to the measured force, 220 pN, and where the shear force approaches -400 pN. The red dashed boxes indicate the same region in **b** and **c** where the dative bond is most likely to be ruptured.

tip-sample interaction energy, whose gradient is proportional to the forces, is negligible (Supplementary Figure 2). The relative frequency shifts from calculations are in excellent agreement with the measured curves for both CO and Cu tips (Fig. 2f, i). To simulate the approach of an AFM tip toward the sample as conducted experimentally, we reduced the tip heights until optimal tip-sample distances are found (see Supplementary Methods, section 3 for details). We first examined the Cu tip (Fig. 3a), which is less complicated than the CO tip due to the multidimensional tilt of the two CO molecules for the CO tip (see Supplementary Figure 3, and Supplementary Data 1-6). We find the vertical force acting on the Cu tip base ($F_z$) is attractive, reaches a maximum of $-156$ pN. This agrees with the experimentally measured force of $-150 \pm 30$ pN very well (Fig. 3b). In addition, a transition from low to high spin occurs when the Fe-C bond is stretched to ~1.9–2.1 Å (shaded area in Fig. 3b), indicating the rupture of the dative bond in CO-FePc reported in a previous study[30]. When applying the same computational method to the CO tip, we find the dative bond in CO-FePc is not broken by decreasing the tip height $z$ when a compressive vertical force ($F_z$) is applied on top of the center of the CO-FePc complex ($x = 0$). In our calculation, we displaced the CO tip horizontally ($x \neq 0$) while decreasing the tip height, similar to the experimental scanning of AFM tip (Fig. 4a). Here,

we performed additional structural relaxation calculations as we moved the CO tip away from the center to ensure our equilibrium structure represents a reasonable geometry (see Supplementary Methods, section 3.2 for details). Significantly, we find that a small increase in the vertical force ($F_z$) acting on the tip apex (Fig. 4b) results in a rapid increase of the lateral (shear) force ($F_x$) acting on the C atom in the CO-FePc complex (Fig. 4c) at low tip heights. However, we believe the dative bond is ruptured before the shear force could reach a few nN. To locate the approximate tip positions where the dative bond is ruptured, we also explored the space around the CO-FePc complex with detailed calculations (dashed curves inside the red boxes). We deduce that the bond is ruptured when the shear force on C reaches about 400 pN by fixing the compressive force ($F_z$) exerted on the tip apex at 220 pN (the experimentally measured force). We find a shear force of 400 pN is a reasonable value, given the fact that the dative bond in CO-FePc is weakened by the Cu substrate due to the trans effect. We conclude that the dative bond is ruptured by a lateral force when a compressive force ($+220$ pN) is applied by the CO-terminated tip.

The calculated spin-polarized local density of states projected onto the center Fe atom of the system shows that the removal of CO results in an increase (from 0.00 to 1.20 $\mu_B$) in the net magnetic moment (Fig. 5a, b). This change in spin state is

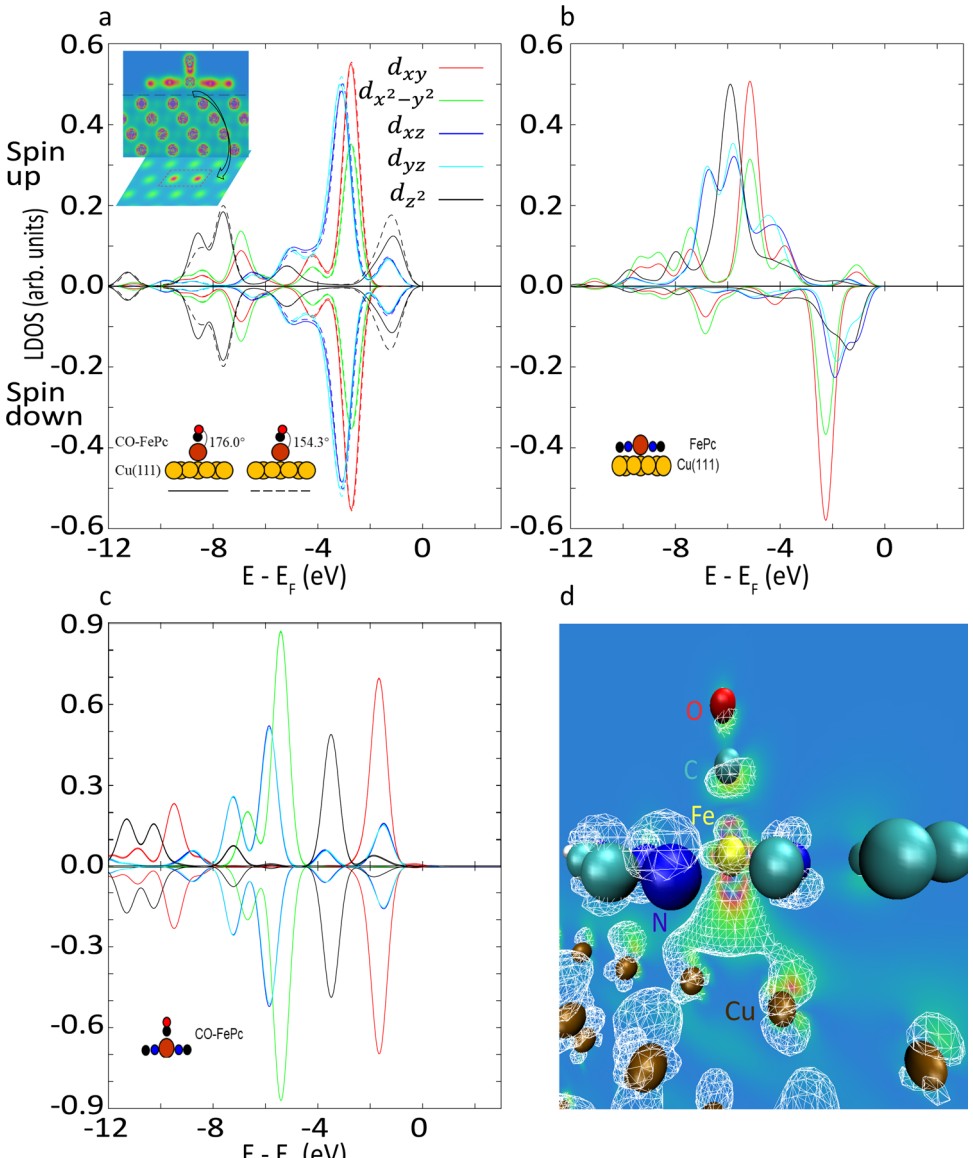

**Fig. 5 The spin-polarized local density of states projected onto the center Fe atom. a** CO-FePc on Cu(111) with two O-C-Fe angles, 176° (at equilibrium, solid curves) and 154° (manually rotated, Fe-C and C=O bond lengths are fixed during rotation, dashed curves), net magnetic moment = 0.0 μ$_B$. **b** FePc on Cu(111), net magnetic moment = 1.2 μ$_B$. **c** CO-FePc without a substrate, net magnetic moment = 0.0 μ$_B$. In **a**, the top inset shows a 2D vertical profile of the total electron density across the Fe atom. The black arrow indicates a 2D horizontal profile in between the complex and the Cu surface where the red dashed rectangle encloses the two bridge Cu atoms underneath the center Fe. **d** The HOMO of the system (around the center Fe atom). A wireframe view of the orbitals is overlapped with a 2D vertical profile of the electron density. Electron density figures are presented in Red-Green-Blue scale (red/blue: higher/lower electron density). Isosurface value = $10^{-5}$ e/bohr$^3$.

consistent with Fig. 3b and a previous DFT study of the CO-FePc complex on the Au(111) surface[30]. The total electron density and the highest occupied molecular orbital (HOMO) of the CO-FePc system demonstrate that the Fe center interacts with the two bridge Cu atoms underneath via the $d_{z^2}$ orbital (Fig. 5a, d). Figure 5a, c shows the presence of the Cu(111) substrate weakens the Fe-C dative bond by shifting the $d_{z^2}$ orbital toward the HOMO, confirming the trans effect by the Cu substrate. To better understand the mechanisms of the bond rupture process, we manually changed the O-C-Fe bond angle by rotating the CO from its equilibrium position (from 176° to 154°) and computed the orbital contribution around the center Fe atom (Fig. 5a). Our choice of the rotated symmetry is not unique, Supplementary Figure 5 shows the results of another bond angle. We find that breaking the almost linear symmetry (176°) of the O-C-Fe bond

shifts the σ-component ($d_{z^2}$) from lower energy states to higher energy states, while changes in the π-components are negligible.

## Discussion

Our measurements show that the dative bond in the CO-FePc complex can be ruptured by either a Cu tip, or by a CO tip. These two tips bring out different mechanisms in bond breaking, i.e., an attractive force of −150 pN by a Cu tip, and a repulsive force of +220 pN by a CO tip. Our real-space DFT calculations accurately predict both the magnitudes and directions of the applied forces exerted on the tip apex ($F_z$). These calculations also reveal that the dative bond is ruptured directly by a vertical force in the case of the Cu tip while the rupture is more likely caused by a lateral shear force when a repulsive force is applied by a CO tip. We also examined the possible influence of different tip geometries,

finding that more complex tips do not affect significantly the tip-sample interaction energy. As for the equilibrium conformation of the CO tip, our calculation is in agreement with previous studies[35,36].

Our calculations confirm that the dative CO-FePc bond is weakened by the presence of the Cu(111) substrate due to the trans effect[29,30]. However, the CO-FePc bond remains a chemical bond, instead of a physisorption[19], based on the computed Fe-C bond length (~1.7 Å). Furthermore, upon the dislodging of the CO by the AFM tip, a ~30 pm downward shift of FePc observed in the experiment revealed that the FePc molecule was lifted ~30 pm upward from Cu(111) surface by attaching a single CO molecule to Fe. This observation is consistent with our calculated shortening of the distance between the Fe atom and the surface of the Cu substrate from 273 to 248 pm, further confirming the rupture of the chemical bond.

When a CO tip is used, the dative bond ruptures at 220 pN, after passing a maximal force of about 300 pN, as shown in Fig. 2g. This trend is consistent with a sequential cleavage of the σ-donation and π-back donation of the dative bond as Fig. 5a shows. The σ-bond has a higher symmetry than the π-bond and hence a higher force is required to tilt and weaken the σ-bond before the dative bond is completely ruptured by the lateral force.

Metal-Pc molecules are widely used model catalysts system for electrochemical reduction of $CO_2$[37,38]. Our study is important for understanding how the activities of Metal-Pc molecules can be manipulated and controlled with single atomic level engineering, and for the design of new FePc-based catalysts. In addition, our results clarify the different mechanisms in bond breaking induced by inert and active tips. This detailed information on the rupture of the bond between CO and FePc will allow us to better understand other dative bonds, such as CO-heme interactions in biochemistry[24,39], as well as chemical reactions of materials under mechanical stress[40,41].

## Methods

**Experimental parameters**. Our experiments were performed with a CreaTec[TM] STM/AFM system under ultrahigh vacuum conditions of ~$10^{-10}$ mbar and a temperature of approximately 5 K. The qPlus sensor has a resonance frequency of 30 kHz with a spring constant $k = 1800$ N/m. In our measurements, the quality factor of the sensor is about 20,000. To minimize crosstalk between the qPlus signal and the STM channel, no voltage was applied on the tip during the force measurement process. The oscillation amplitude was set to be 100 pm.

**Chemicals and sample preparation**. The iron(II) phthalocyanine (FePc, dye content ~90%, Sigma–Aldrich) molecules were evaporated from a silicon chip via direct heating, and the vapor was subsequently deposited on a Cu(111) substrate held at 5 K. The AFM Cu tip apex was functionalized by controlled pickup of a CO molecule from the substrate[1]. All the experiments were conducted using a pure Cu tip or a CO-functionalized tip.

**DFT modelling and computations**. We computed ground state energies using a real-space pseudopotential DFT code, PARSEC[42]. We employed the local density approximation (LDA) by Perdew-Wang (PW92)[43] for the exchange-correlation functional together with Troullier-Martins norm-conserving pseudopotentials[44]. We also tested another exchange-correlation functional by Ceperley-Alder[45], the differences were negligible. In addition, a previous study showed that LDA and the generalized gradient approximation (GGA) gave similar results for properties of FePc and CO-FePc on Au(111)[30]. We employed boundary conditions that require the electron wave functions to vanish outside a spherical or a slab domain, of which the boundary is at least ~300 pm from the outermost atom. We set the distance between neighbor points in the real-space grid to be 16 pm. The density-weighted self-consistent residual error was less than $10^{-4}$ Ry. We employed a finite difference method to approximate the relative frequency shift profiles based on the computed ground state energies across the middle line of the FePc complex, as indicated in Fig. 2f, i at relatively large tip heights. We performed further structural relaxations when the tips were close to the specimen, as the assumption that the movement of the tip had negligible influence on the electronic structure of the specimen may not be valid. We then applied the Hellmann–Feynman theorem to the total ground state energies to compute the net forces acting on each atom. We employed the frozen density embedding theory and a finite difference method for

AFM image simulations[22] (see Supplementary Information—Image Simulations section and Supplementary Fig. 4 for details).

## Data availability
The data supporting our results are available within this article and the Supplementary Information. The Supplementary Information contains a more detailed description of force calculation, structural relaxation, AFM image simulations and tip conformation tests. In addition, we provide the relaxed atomic coordinates of the systems in Supplementary Figure 3 and in Supplementary Data 1-6.

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

## Acknowledgements

The authors gratefully acknowledge Yeju Zhou, Dan Gregory, Michele L. Sarazen, and Guangming Cheng for help with data processing and general discussion. This work was partially supported by ExxonMobil through its membership in the Princeton E-filliates Partnership of the Andlinger Center for Energy and the Environment. This research made use of the Imaging and Analysis Center operated by the Princeton Institute for the Science and Technology of Materials at Princeton University, which is supported in part by the Princeton Center for Complex Materials, a National Science Foundation Materials Research Science and Engineering Center (Grant No. DMR-2011750). D.F. and J.R.C. acknowledge support from the Welch Foundation under grant F-1837 and the U.S. Department of Energy under DOE/DE-FG02-06ER46286. The National Energy Research Scientific Computing (NERSC) and the Texas Advanced Computing Center (TACC) provided computational resources.

## Author contributions

P.C. and N.Y. designed and carried out the experiments. D.F. and J.R.C. performed the DFT calculations. A.S. and E.A.C. provided theoretical insight. P.C., N.Y., Y.Z., D.F., drafted the manuscript with the input from A.S., J.R.C., E.A.C., C.B.A., D.C.D and S.P.R. N.Y. directed the project. All authors discussed the results and contributed to the interpretation and conclusions.

## Competing interests

The authors declare no competing interests.
