## [Peer Review File · Nature Communications]

REVIEWER COMMENTS

Reviewer #1 (Remarks to the Author):

The work by Chen et al. is clearly described and concerns the study of the binding of CO molecules to the centre of a FePc complex adsorbed on a Cu(111) substrate. The experiments are performed using a commercial low-temperature STM/AFM combination based on a qPlus sensor. The experiments are compared with the results of DFT computations that have been specifically performed for the analysis of the experimental data. The results appear correct and can be rationalized by the help of the calculations.

In addition to a few minor points, that I will list below, my main concern is the novelty of this work. Indeed, the authors refer to related work (refs. 17 and 18). The authors continue to claim that "...the controlled breaking of a chemical bond using mechanical forces, along with accurate measurements of these forces, is also important and has not yet been achieved." However, to my knowledge there are several works that report this, see e.g Wagner et al., Phys. Rev. Lett. 109, 076102 (2012), "Measurement of the Binding Energies of the Organic-Metal Perylene-Teracarboxylic-Dianhydride/Au(111) Bonds by Molecular Manipulation Using an Atomic Force Microscope".

Given the fact that quite a few groups have studied bond breaking at surfaces using AFM/STM, the question of which general knowledge is obtained becomes more pressing. What do we really learn from this experiment that we did not already know? Where is it different from previous experiments? The fact that the type of bond is different from that in previous studies is not enough, if we do not find any specific character in the experiments that demonstrates the role of this bond. The main distinction in the present experiment appears to be the application of shear forces for breaking a bond. The authors should clarify more clearly what we learn from this and why this could be important. Which aspects of the experimental and computational results are unexpected or surprising?

Minor points:

1. In Fig. 1, please indicate scale for images b-e. Are b,c and d,e shown on the same scale? (the scale looks different)
2. Caption of Fig. 1 "The minus sign indicates a relative tip height." I did not find a minus sign in the figure.
3. P.5 line 107: What do the authors mean by the word "original"?
4. Details of the deconvolution of frequency shift to force should be given in the Methods.

5. P. 6 line 112: The authors should explain why the force crosses a maximum, and the bond breaks at a lower force. Why is the bond not broken at the maximum force?
6. The simulation in Fig. 2c fails to describe the negative frequency shift. Can the authors explain this?
7. The color scale bar on the right of Fig. 3e has no units. This should probably be pN.
8. The word “contour plot” (caption Fig. 3) is not appropriate for this type of plot.
9. The color scale in Figs. 3d and 3e should be adjusted. Since the relevant process happens at 220pN it would be useful to adjust the color scale such that the contour of 220pN does not lie in a homogeneously blue area.
10. The “red dashed rectangle” (caption Fig. 4) is not visible.
11. The oscillation amplitude of the qPlus sensor was set to 100pm. This may not be negligible. The authors should discuss the effect that this amplitude may have on their measurement results.

Reviewer #2 (Remarks to the Author):

The bond between CO and a metal atom is very important in atomic force microscopy, as CO terminated tips mounted on quartz cantilevers are the method of choice to image organic molecules and metal clusters with atomic resolution.

Chen et al demonstrate that CO bonded to an FePc molecule can be broken off both by a reactive metal tip as well as by a CO terminated metal tip. DFT calculations support the experimental findings in a very convincing way. A threshold of 150 pN in pulling and 220 pN in pushing is required to break off the CO. These findings explain, why CO tips can still be used for atomic manipulation as demonstrated by Berwanger (PRB 2018). Thus, the findings presented are an important contribution to the current state of science and well suited for publication in Nature Communications.

I do have a few comments that should be considered before the paper can be published.

1. The introduction claims that "subatomic" resolution has been obtained with a CO tip by Gross in 2009 - this should be corrected to "atomic" resolution. Subatomic resolution with CO tips has been demonstrated by Huber in ref. 18.
2. Units are missing in the color bar right of Fig. 3e (piconewtons??)
3. What does (vide infra) mean in the paragraph after Fig. 3?
4. Berwanger studied the reactivity of iron clusters by CO tips in PRL 124 096001 and found that CO tips can exert attractive short range forces of 400 pN without falling off. Is that compatible with the findings presented here? Does it mean, CO can stick better to a solid metal tip than to a single Fe atom on the FePc molecule?
5. The final paragraph in "Discussion and conclusion" is hard to comprehend, I suggest to rephrase it and perhaps lengthen it a bit to lay out the important main findings in a more comprehensible fashion.

Reviewer #3 (Remarks to the Author):

This work reported detailed bond rupture process of the dative bond between a carbon monoxide molecule and a ferrous phthalocyanine (FePc) by AFM tip-induced mechanical forces. The process was quantitatively measured and characterized both experimentally and through quantum mechanical DFT calculations. CO-FePc bond rupture was realized by either attractive forces induced by Cu-tip or shear forces produced by compression of CO functionalized Cu-tip. The presentation is concise and clear, revealing detailed subatomic resolution of bond breaking in a simple molecular complex.

While I have no doubt about the STM/AFM imaging and force measurements, my major concern is the interpretation of the experimental results. Atomic conformations shown in Figs. 2a & 3a-c are clearly of very high energy states, whereas the actual tip geometry and contact are far from being properly characterized. This is a challenging issue in the STM/AFM community. My suggestion is in the SI, both Cu tip and CO modified tip geometries and their variations should be clearly presented.

The optimized Cu₂ cluster aligned vertically along the center axis of the molecule is certainly not a realistic case. CO-modified tip is even more challenging (SI page 3 - 4). However, at least different variations of the tip contact geometries, especially the equilibrium conformation of CO on the tip should be sufficiently discussed. More importantly, in Fig. S1, all the conformations of tip-molecule-substrate should be clearly shown in the figure.

An interesting question, which I am not sure if the authors could properly address, is whether there is a rate issue when measuring the bond breaking forces (either under compression or attraction).

A point-by-point responses to the reviewers' comments are detailed below. The reviewer's comments are given in black and our responses are in blue:

REVIEWER COMMENTS

Reviewer #1 (Remarks to the Author):

The work by Chen et al. is clearly described and concerns the study of the binding of CO molecules to the centre of a FePc complex adsorbed on a Cu(111) substrate. The experiments are performed using a commercial low-temperature STM/AFM combination based on a qPlus sensor. The experiments are compared with the results of DFT computations that have been specifically performed for the analysis of the experimental data. The results appear correct and can be rationalized by the help of the calculations.

We thank the reviewer for these positive comments.

In addition to a few minor points, that I will list below, my main concern is the novelty of this work. Indeed, the authors refer to related work (refs. 17 and 18).

We agree with the reviewer that novelty should be a major criterion for publication. We believe that our study fully satisfies this criterion as detailed below.

Our findings are clearly different from previous studies. In reference 18 (*Sci. Adv.* **3**, 2017, previously Ref. 17). Kawai et al. measured the strength of the C=O...H-C hydrogen bond between a CO-tip and an aromatic hydrocarbon with vertical C-H bonds (see figure below). The focus of that paper was on imaging characterization of H atoms and very weak hydrogen bonding before the onset of Pauli repulsion. This result is very different from ours. Our focus is centered on the rupture of a well-defined chemical bond (CO-FePc). Still, this reference is relatively close to our study and is why we cite it as a key reference.

From Figure 3A in Ref. 18 (*Sci. Adv.* **3**, 2017, previously Ref. 17)

In reference 19 (*Science* **366**, 2019, previously Ref. 18), Huber et al. investigated the formation of a single chemical bond following the transition from physisorption to chemisorption. Instead, our paper focuses on the controlled rupture of a chemical bond.

Our work builds upon these previous results and represents a significant step forward in understanding the bond breaking process.

From Figure 1 in Ref. 19 by Huber et al. “Chemical bond formation showing a transition from physisorption to chemisorption”. (*Science* **366**, 2019, previously Ref. 18)

The authors continue to claim that “...the controlled breaking of a chemical bond using mechanical forces, along with accurate measurements of these forces, is also important and has not yet been achieved.” However, to my knowledge there are several works that report this, see e.g Wagner et al., *Phys. Rev. Lett.* **109**, 076102 (2012), “Measurement of the Binding Energies of the Organic-Metal Perylene-Teracarboxylic-Dianhydride/Au(111) Bonds by Molecular Manipulation Using an Atomic Force Microscope.”

We modified the sentence “the controlled breaking of a chemical bond using mechanical forces, along with accurate measurements of these forces, is also important and has not yet been achieved” to read “the controlled breaking of a chemical bond using mechanical forces, along with accurate measurements of these forces, is also important although its detailed process has not yet been fully understood”.

Following the reviewer’s comments, we carefully examined the paper by Wagner *et al.* (*Phys. Rev. Lett.* **109**, 076102 (2012)). In that study, a sheet of (or planar) “PTCDA molecule is repeatedly detached from the metal surface” and the “molecule is strongly distorted during the lift-off” (see Figure 1 below taken this paper). Although the authors experimentally determined the individual binding energy contributions to an organic-metal bond, the interactions (van der Waals interactions, metal-molecule hybridization, and local chemical bonds) are non-specific. Hence, they are very different from the rupture of a well-defined single chemical bond in CO-FePc as reported in our study.

From Figure 1 in “Measurement of the Binding Energies of the Organic-Metal Perylene-Tetracarboxylic-Dianhydride/Au(111) Bonds by Molecular Manipulation Using an Atomic Force Microscope.” C. Wagner, N. Fournier, F. S. Tautz, and R. Temirov. *Phys. Rev. Lett.* **109**, 076102 (2012).

Given the fact that quite a few groups have studied bond breaking at surfaces using AFM/STM, the question of which general knowledge is obtained becomes more pressing. What do we really learn from this experiment that we did not already know? Where is it different from previous experiments? The fact that the type of bond is different from that in previous studies is not enough, if we do not find any specific character in the experiments that demonstrates the role of this bond. The main distinction in the present experiment appears to be the application of shear forces for breaking a bond. The authors should clarify more clearly what we learn from this and why this could be important. Which aspects of the experimental and computational results are unexpected or surprising?

As pointed out and appreciated by the reviewer, one important distinction of our work relative to previous studies is the use of a shear force for breaking a chemical bond. The contribution of the shear force was actually unexpected, and we identified it only after detailed theoretical modeling of our experimental measurements. By combining quantitative experiments with modeling, we also obtained the following new results:

- (1) The same dative bond (CO-FePc) was ruptured by both a CO-tip and a Cu-tip. These two tips emphasize different mechanisms in bond breaking, *i.e.*, a repulsive force of +220 pN by a CO-tip, and an attractive force of -150 pN by a Cu-tip.
- (2) Our computational results are in excellent agreement with the measured forces and characteristics of CO- and Cu-tips. More importantly, they provide a deeper understanding of the complex interactions between the tip, CO, FePc and metal substrate that cannot be directly measured by experiments. For example, the computed forces acting on each atom revealed that the measured force corresponds to the vertical force acting on the AFM tip apex.
- (3) A coherent picture and further understanding of the experimental details were obtained. The dative CO-FePc bond weakens due to the “ π back donation” by the underneath Cu substrate. With a CO-tip, a force lower than the maximal force ruptured the dative bond. We propose that the larger force is required to first tilt the stronger σ -bond before the dative bond is fully broken. To confirm our hypothesis, we performed new calculations in which we manually rotated the decorated CO by 7.2° (Fig.S5) and 21.7° (Fig. 4a, see below), and then determined the orbital contribution around the center

Fe atom. We found that the d_z^2 component of the σ -bond shifted toward the HOMO while the changes in the π components were negligible. As a result, cleavage of the C-Fe bond can be expected to occur as a two-step sequential process where a higher force is first required to tilt and weaken the σ -bond before the dative bond is completely ruptured by the lateral force.

Fig. 4a from the main text.

- (4) Our study shows that a dative bond (CO-FePc) can be readily ruptured by a CO-tip and demonstrates that a CO molecule can stick better to a solid Cu tip than to the single Fe(II) center of the FePc molecule. This finding explains why CO tips can still be used for atomic manipulation as demonstrated by Berwanger, *et al.* (*PRB* **98**, 195409 (2018)). Our finding is also supported by a recent report (*PRL* **124**, 096001 (2020)) that CO tips can exert attractive short-range forces up to 400 pN.
- (5) We demonstrated that a FePc molecule can be lifted ~ 30 pm upward from Cu(111) surface by attaching a single CO molecule to the Fe. Single-atom catalysis is a new frontier in heterogeneous catalysis (*Nat Rev Chem* **2**, 65 (2018)). The single-site catalysis community continues to explore sensitive detection metrics (*Chem. Rev.* **113**, 10, 8152 (2013)). Thus, an accurate determination of the molecule position and force required to break a dative bond is important. Metal-Pc molecules are widely used catalysts for the oxygen reduction reaction. As a study of a model system, our results will be very valuable for understanding how activities can be manipulated and controlled with single-atom engineering, and for the design of new FePc-based catalysts (*J. Am. Chem. Soc.* **143**, 2, 925 (2021)).
- (6) The “magic-angle” exotic phenomena discovered in the twisted bilayer graphene is mainly attributed to the modification of the local environment using controlled regulation of local electron-electron interactions in a two-dimensional fashion (*Nature* **556**, 43 (2018)). Our demonstration of using various AFM tips to introduce changes of the spin state at specific atoms could potentially introduce new physical and chemical properties at the level of a single atom.

In summary, we believe that our results are novel and significant. Building upon previous studies, we achieved a better level of understanding of the single bond breaking process. The other reviewers have also highlighted the novel aspects of our work.

Based on this reviewer's comments, we also introduced the following changes to the manuscript:

(1) In the abstract, we have included a sentence on the contribution of shear forces on the breaking of the chemical bond.

(2) We have included the paper by Wagner et al. (*Phys. Rev. Lett.* **109**, 076102 (2012)) as suggested by the reviewer. (Ref. 17)

Minor points:

1. In Fig. 1, please indicate scale for images b-e. Are b,c and d,e shown on the same scale? (the scale looks different)

The image size of b,c is 2 nm by 2 nm, and d,e is 1.33 nm by 1.33 nm. Scale bars were added to the updated figure. In addition, the color scale of the simulation images was adjusted as an insert in d. The scale now runs from -22 to -12 (arbitrary units), which is consistent with the experimental image.

2. Caption of Fig. 1 "The minus sign indicates a relative tip height." I did not find a minus sign in the figure.

We placed the minus sign in the caption, rather than in the figure.

3. P.5 line 107: What do the authors mean by the word "original"?

We deleted the word "original".

4. Details of the deconvolution of frequency shift to force should be given in the Methods.

The deconvolution from the measured frequency shift to energy and force curves is discussed in detail in previous papers, (*APL*, 84, 1801 (2014)).

Irrespective of the amplitude of oscillation, the formulas for the force and energy in terms of measured frequency shift we adopted are:

$$F(z) = 2k \int_z^\infty \left(1 + \frac{a^{1/2}}{8\sqrt{\pi(t-z)}} \right) \Omega(t) - \frac{a^{3/2}}{\sqrt{2(t-z)}} \frac{d\Omega(t)}{dt} dt \quad (1)$$

$$U(z) = 2k \int_z^\infty \Omega(t) \left((t-z) + \frac{a^{1/2}}{4} \sqrt{\frac{t-z}{\pi}} + \frac{a^{3/2}}{\sqrt{2(t-z)}} \right) dt \quad (2)$$

where $\Omega(z) = \Delta\omega(z)/\omega_{\text{res}}$, k is the spring constant of the qPlus sensor and a is the tip oscillation amplitude.

The accuracy and validity of these formulas were also demonstrated by a simulated experiment in *APL*, 84, 1801 (2014).

In our measurements, the oscillation amplitude of the tip is 100 pm. The spring constant of qPlus sensor is 1800N/m.

5. P. 6 line 112: The authors should explain why the force crosses a maximum, and the bond breaks at a lower force. Why is the bond not broken at the maximum force?

In the discussion section of the manuscript, we pointed out that the lateral force tilts the dative bond of CO-FePc. This tilting occurs prior to the rupture of the bond, though it is unfavorable due to the symmetry of σ donation. This is consistent with the fact that right before the bond breaking a higher force is observed in the experiment (Fig. 2d), which is big enough to break the symmetry of the bond and therefore weaken the bond. Additional discussion is highlighted in the paragraph for new result (3) on Page 4.

6. The simulation in Fig. 2c fails to describe the negative frequency shift. Can the authors explain this?

The negative frequency is likely caused by the fixed scanning geometry in our simulation at large tip heights. We place the tip at a relatively large height so that the geometry changes within both the tip and the sample can be neglected. In the experiment, especially at small tip heights, when the CO tip gets closer to the CO, it slightly tilts. The Figure below shows that the calculated energy, force and df curves have similar shapes. Relaxing the entire system during scanning will result in lower energies near the center as the tip is allowed to tilt. As the red arrows indicate, if extra structural relaxation is performed, we expect to obtain an energy curve with a larger curvature, which will then result in a df curve that is closer to the experimental curves. For the Cu tip, in Fig.2f, the simulated curve matches the experimental curves very well. This is because the metal Cu tip is more rigid so that our approximation is very accurate.

DFT computed energy (Ry) curve. Calculated vertical force (pN) and the relative df curves by taking numerical derivatives of the energy data.

Performing additional structural relaxation at large tip heights at each grid point is computationally intensive, and unlikely to add substantive content to our conclusions. Nevertheless, to ensure the force calculations are accurate, we performed structural relaxation at each grid point when the tip is close to the sample; as shown in Fig. 3, we optimized the entire system (tips+sample+surface) in the calculations because it is near the breaking point.

7. The color scale bar on the right of Fig. 3e has no units. This should probably be pN.

This has been fixed in the updated figure.

8. The word “contour plot” (caption Fig. 3) is not appropriate for this type of plot.

We changed the word “contour plot” to “line cuts of 3D force maps”.

9. The color scale in Figs. 3d and 3e should be adjusted. Since the relevant process happens at 220 pN it would be useful to adjust the color scale such that the contour of 220pN does not lie in a homogeneously blue area.

The color scale was chosen with the purpose to show that when the measured vertical force (Fig. 3d) surpasses 220 pN, the shear force acting on the bottom C atom (Fig. 3e) can increase rapidly to over 1000 pN (the “yellow-red” regime). If we adjust the color scale to “0 – 500 pN” or something close, we will not be able to see how the shear force changes in the bottom of Fig. 3e (see the Figs. X and Y below).

Fig. X

Fig. Y

10. The “red dashed rectangle” (caption Fig. 4) is not visible.

This has been fixed.

11. The oscillation amplitude of the qPlus sensor was set to 100pm. This may not be negligible. The authors should discuss the effect that this amplitude may have on their measurement results.

In general, the oscillation amplitude will not affect the measurement results. This is because the frequency shift signal is a convolution of a semispherical weight function with the tip-sample force gradient. The radius A of the weight function is equal to the oscillation amplitude of the cantilever (*Rev. Mod. Phys.* 75, 949 (2003)). We work with a small amplitude of about 100 pm to probe the short-range force, such as the Pauli repulsion force. Given that the bond rupture scale probed here is comparable to short-range force scale, a 100 pm tip amplitude can accurately probe the interaction. In our experiment, a change in the tip oscillation amplitude from 50 pm to 200 pm was tested. We did not observe any effects on the measured force value, although decreasing the amplitude will increase the system noise. Increasing the amplitude could decrease the measured noise but also could decrease the weight function of the short-range force and blur the AFM images. For this reason, we usually choose 100 pm tip oscillation amplitude for both imaging and force measurement.

Reviewer #2 (Remarks to the Author):

The bond between CO and a metal atom is very important in atomic force microscopy, as CO terminated tips mounted on quartz cantilevers are the method of choice to image organic molecules and metal clusters with atomic resolution.

Chen et al demonstrate that CO bonded to an FePc molecule can be broken off both by a reactive metal tip as well as by a CO terminated metal tip. DFT calculations support the experimental findings in a very convincing way. A threshold of 150 pN in pulling and 220 pN in pushing is required to break off the CO. These findings explain, why CO tips can still be used for atomic manipulation as demonstrated by Berwanger (PRB 2018). Thus, the findings presented are an important contribution to the current state of science and well suited for publication in Nature Communications.

We thank the Reviewer for their positive assessment. We added the suggested paper (PRB 98, 195409 (2018)) as Ref. 34.

I do have a few comments that should be considered before the paper can be published.

1. The introduction claims that "subatomic" resolution has been obtained with a CO tip by Gross in 2009 - this should be corrected to "atomic" resolution. Subatomic resolution with CO tips has been demonstrated by Huber in ref. 18.

We have changed the word "subatomic" to "atomic".

2. Units are missing in the color bar right of Fig. 3e (piconewtons??)

We have now included the units (pN) in Fig. 3e.

3. What does (vide infra) mean in the paragraph after Fig. 3?

"(vide infra)", from Latin, means "see below". We deleted the phrase.

4. Berwanger studied the reactivity of iron clusters by CO tips in PRL 124 096001 and found that CO tips can exert attractive short range forces of 400 pN without falling off. Is that compatible with the findings presented here? Does it mean, CO can stick better to a solid metal tip than to a single Fe atom on the FePc molecule?

We thank the reviewer for pointing this out. Yes, our findings have shown that CO on the Cu tip is more stable than that on the FePc molecules. Neither the 150 pN attractive force nor the 220 pN repulsive force can break the CO-tip bonding. We have included this paper (PRL 124 096001 (2020)) in the reference list.

5. The final paragraph in "Discussion and conclusion" is hard to comprehend, I suggest to rephrase it and perhaps lengthen it a bit to lay out the important main findings in a more comprehensible fashion.

We reworked the final paragraph in "Discussion and conclusion" according to the reviewer's suggestion.

Reviewer #3 (Remarks to the Author):

This work reported detailed bond rupture process of the dative bond between a carbon monoxide molecule and a ferrous phthalocyanine (FePc) by AFM tip-induced mechanical forces. The process was quantitatively measured and characterized both experimentally and through quantum mechanical DFT calculations. CO-FePc bond rupture was realized by either attractive forces induced by Cu-tip or shear forces produced by compression of CO functionalized Cu-tip. The presentation is concise and clear, revealing detailed subatomic resolution of bond breaking in a simple molecular complex.

We thank the reviewer for their positive comments.

While I have no doubt about the STM/AFM imaging and force measurements, my major concern is the interpretation of the experimental results. Atomic conformations shown in Figs. 2a & 3a-c are clearly of very high energy states, whereas the actual tip geometry and contact are far from being properly characterized. This is a challenging issue in the STM/AFM community. My suggestion is in the SI, both Cu tip and CO modified tip geometries and their variations should be clearly presented. The optimized Cu₂ cluster aligned vertically along the center axis of the molecule is certainly not a realistic case. CO-modified tip is even more challenging (SI page 3 - 4). However, at least different variations of the tip contact geometries, especially the equilibrium conformation of CO on the tip should be sufficiently discussed.

As the reviewer pointed out, the actual AFM tip geometry and contact have been challenging issues in the STM/AFM community, see e.g. *APL* 114, 143103 (2019), and *Phys. Rev. Research* 2, 033094 (2020). Following the reviewer's recommendation, we have tested the effects of using different tip contact geometries, especially for the equilibrium conformation of CO on the tip, in our modeling studies. The interactions between the tips and CO-FePc were also extensively studied. For the Cu tip, we examined Cu₂ and Cu₄ tips; for the CO tip, we tested CO, CuCO, Cu₂CO and Cu₄CO. Before going into the details of these studies, we briefly discuss the models used in the original version of our manuscript. We address why a good AFM Cu tip for high-quality imaging and spectroscopy experiments requires only one Cu atom at the apex tip.

Cu tip. A good AFM tip means that there is no double-tip, tilted-tip, or tip with other geometries. If two atoms are side-by-side at the tip apex (double-tip), we should see duplication of any atomic features. With a tilted-tip, we could see an asymmetrical shadow on one side of all the resolved objects on the surface. These "bad tips" can be easily identified and excluded from the experiment. For symmetric tip geometries, we are able to tell whether the tip is sharp or not. Only a sharp tip can provide clear atomic images, which is the starting point of all our measurements. Otherwise, the atomic images cannot be repeated and the location of the measured molecules or atoms cannot be assigned precisely. Based on these considerations, we are confident that a good tip is characterized by having a single atom dominate the tip apex.

In the simulation, we modeled the metal Cu tip using a Cu₂ tip (two atoms connected vertically) since some of the present authors have shown previously that a more complex tip geometry, such as a Cu₄, does not make any significant difference for the tip-sample interaction (tests were done on Cu₂N and graphene, (*PRB*, 95, 081401(2017))). We assume the Cu tip is rigid so that the tip tilting effect is negligible. Fig. 2f shows that our calculated curve accurately matches the experimental curve. Therefore, we believe our approximation is well justified.

CO tip. In our experiment, in order to achieve the best image quality, we need to select a good metal tip first and then use it to pick up a CO. As for tip modeling in our simulation, previous work by some of the present authors (*Nano Lett.* 16, 3242 (2016)) showed that using a Cu₂CO tip instead of a CO tip does not result in a significant difference in terms of tip-sample interaction energy (tests were done on different sites of a benzene molecule, see the "Figure S1" below). In addition, another of our previous studies shows that using a Cu₂ tip instead of a Cu₄ tip makes no difference in terms of the tip-sample interaction energy (*Phys. Rev. B* 95, 081401(R) (2017)).

Figure S1: Interaction energy curves between CO and benzene (red lines) and Cu_2CO (green lines). Squares, circles and triangles represent hollow (h), carbon (C), and hydrogen (H) sites, respectively.

Additional Tests. Following the reviewer's suggestion, we carried out additional simulations on a CO-FePc molecule using different tips. For the Cu tip, we examined Cu_2 and Cu_4 tips (Fig. S2 a); for the CO-terminated tip, we tested CO, CuCO, Cu_2CO and Cu_4CO (Fig. S2 b). In both cases, we do not see a significant variation in the interaction energy curves vs tip height, indicating that our original models (Cu_2 and CuCO) are meaningful. As shown in Fig. S2b, the interaction energy curves for the different models are very similar.

It is also worth noting that Berwanger *et al.* (*Phys. Rev. Lett.* **124**, 096001 (2020)) recently studied the chemical reactivity of small Fe clusters using a CO tip. They were able to obtain a satisfactory understanding of their experimental results even though they only included a single CO molecule to model the tip in their DFT calculations.

Fig S2. Interaction energy between tip and CO-FePc. **a** Cu₂ and Cu₄ tips; **b** CO, CuCO, Cu₂CO and Cu₄CO tips. All the tips are placed above the decorated CO of CO-FePc. The Cu(111) substrate was not included in the calculations.

Equilibrium conformation of CO. To make sure that the CO tip has the correct equilibrium geometry, we performed structural relaxation calculations as the CO tip approaches the center of the sample molecule. As shown in Fig. S3, our calculation agrees with a previous study by Salmeron (*Science* **343**, 1083 (2014)) on the CO-CO interaction (Fig. MS). As the CO tip approaches the center of the FePc molecule, the two CO molecules bend due to repulsion, with the same (parallel) general orientation as when they bend due to attraction.

We included all the atomic coordinates of the equilibrium conformations of the tip-sample system at different positions in Tables S1 – S6.

Fig S3. (Color online) Relaxed conformations of the Cu apex + CO tip + CO-FePc system at different tip-sample separation distances (left: top view; right: side view). The horizontal distances between the Cu apex and the center Fe atom are: (a) 382 pm, (b) 318 pm, (c) 255 pm, (d) 191 pm, (e) 127 pm and (f) 64 pm. The relaxed atomic coordinates are given in Table S1-S6.

More importantly, in Fig. S1, all the conformations of tip-molecule-substrate should be clearly shown in the figure.

Besides providing the details (bond length, bond angle) of the geometries, we have added Fig. S1e to show how the tip-molecule-substrate system looks like (both top and side view) in our model.

Fig. S1

An interesting question, which I am not sure if the authors could properly address, is whether there is a rate issue when measuring the bond breaking forces (either under compression or attraction).

In our measurements, the tip is scanned very slowly under near-equilibrium conditions, about 100 ms for each point (the scan time in our measurement can only be adjusted in a small range between 20 ms to 100 ms due to the noise and drift limit of the SPM system). We did not see any rate issue when measuring the bond breaking forces because of a change of the scan speed with either the CO or the Cu tip. Using force spectroscopy measurements by AFM or elastic scanning tunneling spectroscopy (STS) measurements by STM, we identify excited or charged molecular states. For example, if the molecules are in a charged state, the local contact potential difference will shift to positive or negative voltage direction in $df(V)$ measurements, depending on the charge state of the molecules (*Nature Nano*, **7**, 227 (2012)). With STS measurements, the change from a ground state to a charged or high-energy state of a molecule can also be resolved (*Science* **23**, 5683 (2004)).

A list of changes we made in Figures:

1. Fig. 1, we added a color bar.
2. Fig. 1, we added scale bars.
3. Fig. 1, we adjusted the contrast of the simulated images (**d** and **e**).
4. Fig. 2, we added two red arrows indicating the break point in **d** and **g**.

5. Fig. 3, we added unit “(pN)” above the color bar of **e**.
6. Fig. 3, we improved the quality/resolution of the figure (fixed the red boxes and removed the white bar)
7. Fig. 4, we added the x and y coordinates for **a**, **b** and **c**.
8. Fig. 4, we added the calculation results for CO-FePc on Cu(111) with another O-C-Fe bond angle (154.3°) to explain the sequential bond cleavage of the dative bond (dashed curves in **a**).
9. Fig. S1, we added **e** to show the entire system (Cu apex + CO tip + CO-FePc complex + 4 layers of Cu(111) substrate) in our simulation.
10. Fig. S2 (new Figure) illustrates the effect of using different tip geometries in our simulation.
11. Fig. S3 (new Figure) shows a dynamic process of tip approaching the center of the CO-FePc molecule horizontally. The relaxed atomic coordinates of all the geometries are given in Table S1-S6.
12. Fig. S4 (was Fig.S2). No change made.
13. Fig. S5 (new Figure) shows an additional tilting angle we tested. It is supplemental to Fig.4**a**.

REVIEWERS' COMMENTS

Reviewer #1 (Remarks to the Author):

The authors have made an extensive revision of their manuscript and clarified more explicitly why new information is obtained from their results. The revised paper can be accepted for publication.

Reviewer #2 (Remarks to the Author):

Thank you for addressing all my queries in a satisfactory way. In my view, the paper is fit for publication now.

Reviewer #3 (Remarks to the Author):

The revised version satisfactorily answered most of my questions. However,

- 1) Given the nominal tip apex radius is usually around 10 nm (more or less), it is very likely that the tip apex should be a protrusion / atomic step on which CO is sitting (in fact normal standing) on it. I suggest authors specifically make this clear, rather than test Cu₂/Cu₄ cluster (though these new calculations are helpful);
- 2) When talking about 20 ms - 100 ms for each point, what is the actual scan speed (or approach speed) between CO - CO molecules in nm/s?
- 3) To increase the impact of this work, especially for computational atomic simulation community to understand the imaging and force spectroscopy, the actual normal and lateral spring force constants of the STM/AFM cantilever should be clearly described (in N/m). Consequently, in SM Fig S4, the lateral force constant of CO tip taken as an adjustable parameter, which depends on the tip height, is quite confusing. This has to be made clear.

REVIEWERS' COMMENTS

Reviewer #3 (Remarks to the Author):

The revised version satisfactorily answered most of my questions. However,

1) Given the nominal tip apex radius is usually around 10 nm (more or less), it is very likely that the tip apex should be a protrusion / atomic step on which CO is sitting (in fact normal standing) on it. I suggest authors specifically make this clear, rather than test Cu₂/Cu₄ cluster (though these new calculations are helpful);

Response:

We understand that the nominal tip apex radius is significantly larger than our model. However, it is not possible to model the entire tip in simulation. Therefore, we started from the simplest case. For example, to model the CO-functionalized tip, we started from a single CO molecule, followed by CuCO, Cu₂CO and Cu₄CO. We found the effect of including more Cu atoms is negligible (see the figure below). We then assumed that this conclusion can be extended to larger systems (e.g., a tip apex with a radius of around 10 nm). In addition, there are some other groups in this field obtained reliable results using similar models (either a single CO or a CO with a small Cu cluster).

In Supporting Information – section (3) Probe tips with CO-FePc complex on a Cu(111) substrate, we modified the text to address this issue.

“As for probe tip modeling, while some groups have modeled the probe tip as a combination of a metal cluster with an apex functionalized tip⁷⁻⁹, we obtain accurate images without including the metal cluster¹⁰⁻¹¹ despite the fact that the nominal tip apex radius is significantly larger than these theoretical models.”

2) When talking about 20 ms - 100 ms for each point, what is the actual scan speed (or approach speed) between CO - CO molecules in nm/s?

Response:

We added a subsection named “AFM tip scan speed” to address this issue in the SI (see below):

“For AFM measurements, the noise level can be adjusted by the scan speed of the tip. The longer the tip stays on one point, the lower the noise is. However, increasing the time of each point will also increase the drift, which will result in distortion of the image and inaccuracy of the data.

The scan speed is about 20 ms - 100 ms for each point, so the actual scan speed during our measurement is about 0.5 nm/s.”

3) To increase the impact of this work, especially for computational atomic simulation community to understand the imaging and force spectroscopy, the actual normal and lateral spring force constants of the STM/AFM cantilever should be clearly described (in N/m). Consequently, in SM Fig S4, the lateral force constant of CO tip taken as an adjustable parameter, which depends on the tip height, is quite confusing. This has to be made clear.

Response:

First, the lateral spring constant of the CO tip, k_{CO} , is an *intrinsic parameter*. In our previous work on carbon nanotubes (*Phys. Rev. B* **58**, 12649 (1998)), we find the spring constant to be essentially independent of diameter and number of layers of carbon nanotubes. Here, k_{CO} was experimentally determined to be around 0.24 N/m in previous work (*Science* **343**, 1120-1122 (2014)). They used a CO-terminated tip to probe a single CO molecule on a Cu(111) substrate (see the figure below). In our work, we assume a simple linear relationship between the lateral force and the displacement ($\vec{\Delta}_{lat}(x, y) = \frac{\vec{F}_{lat}(x, y)}{k_{CO}}$); different values of k_{CO} were tested in order to achieve closer agreement with the experimental images (0.4 N/m, 0.8 N/m, etc.).

Second, k_{CO} does not depend on the tip height. In the simulation, we treat it as an *adjustable parameter* which represents the lateral stiffness of the tip. Fig. S4 (renamed as Fig. 4 in the SI) illustrates the effect of using different values of k_{CO} . In the main text, we set k_{CO} to be 0.80 N/m to achieve closer agreement with the experimental images.

Finally, in the SI, we added the following sentence below Eq.2 to make this clear:

“ k_{CO} is an adjustable parameter set to 0.80 N/m in the main text to achieve closer agreement with the experimental images.”